# PERIODNET: LIGHTWEIGHT AND EFFICIENT TIME SERIES PREDICTION MODEL BASED ON PERIODIC CHARACTERISTICS

## ABSTRACT

The task of multivariate time series prediction has always been a challenging task. In this field, various related methods emerge in endlessly, whether based on fully connected, convolutional neural networks or attention-based models, all have achieved remarkable results. However, current long-term prediction tasks mainly rely on complex attention mechanisms or causal convolutions, which result in huge computational costs and are not suitable for edge devices or scenarios with limited computing resources. Therefore, our research focuses on lightweight time series prediction model exploration. Our main work focuses on the analysis of time series data, focusing on the importance of periodic features and the fusion of local features and global features. Based on the mathematical idea of Fourier series, we designed a simple and lightweight module for extracting periodic features; and designed a lightweight module that can effectively fuse local information and global information, thereby enhancing Feature representation and prediction performance. By comparing with the current state-of-the-art results, we verified the effectiveness of the module we designed. On 7 benchmark data sets including etth1, etth2 and ili etc., our model achieved significant performance improvements compared to the state-of-the-art results. The specific code of our research results can be found at https://github.com/sep21Be/periodNet.

## 1 INTRODUCTION

Multivariate time-series prediction tasks have a wide range of application prospects at present, which can be applied to many fields, including weather prediction, financial prediction, epidemic development prediction and so on. However, various neural network models for prediction are currently overly complex with numerous parameters, such as the model based on the attention mechanism (Vaswani et al. (2017)), which constructs the Transformer by stacking the attention mechanism, demonstrates excellent generalization ability, and achieves excellent results in numerous fields, which also include the time-series prediction task (Li et al. (2019); Zhou et al. (2021); Wu et al. (2021); Liu et al. (2021); Zhou et al. (2022); Chen et al. (2022); Nie et al. (2022), Zhang & Yan (2022)). Unfortunately, although the model based on the attention mechanism performs well, it requires the computation of the attention matrix, which necessitates the sacrifice of a large amount of memory and computational resources, as well as the addition of extra layers by which to provide position information.

On the other hand, classical TCN models(Bai et al. (2018)), i.e., causal convolutional network-based models, are designed specifically for time-series data, where the model only has access to historical data, similar to an attention mechanism that incorporates an attention mask, i.e., $x_n$, which can only observe data up to $x_{n-1}$, but not $x_{n+1}$ and beyond.In addition, causal convolution usually requires the use of longer convolution kernels to take into account historical information, which means that at each time step, more convolution operations need to be computed. And in many applications causal convolution is stacked into multiple layers in order to capture features at different time scales. For these reasons, models that employ causal convolution tend to require a lot of memory and computational power, but at the same time, the complex design and the large number of computations have led to fairly good results for models based on causal convolution.

In recent research(Nie et al. (2022)), by decomposing the time series sequence and then connecting it with a fully connected layer, or directly using a layer of fully connected layers, the effect has surpassed many models based on Transformer and TCN.

Based on the above, we can't help but think which is more important in time-series prediction, trend or period? Can a lightweight model with fewer parameters beat a transformer-based model with more parameters? Maybe we need to do more feature-related work in time-series prediction.

Therefore, we propose a model based on the mathematical idea of Fourier series and the fusion of local and global features, focusing on time series decomposition to extract periodic features and enhance the decomposed features. The model has the following features:

- stronger parsing, we adopt the mathematical idea of Fourier series, explicitly periodical feature extraction and periodical data decomposition from time series data.
- lower number of parameters, the model uses basic deep learning modules, including full connectivity, average pooling and convolution, instead of the currently popular Transformer architecture. This results in a more lightweight model suitable for resource-constrained application scenarios.

## 2 RELATED WORK

In the field of time series decomposition, there are a number of classical works(Holt (1957); Cleveland et al. (1990); Vautard et al. (1992); Kalekar et al. (2004); De Livera et al. (2011)) that have achieved superior results in the past non-deep learning era and have relatively strong interpretability of decomposition results. However, such methods have a relatively large drawback of being computed by artificially defined operators, which brings cumbersome computation and at the same time cannot be adapted to most cases. Recently, the field of deep learning has begun to introduce the decomposition idea of the classical method of time series decomposition(Zeng et al. (2023); Wang et al. (2022)), through the data-driven way to calculate the decomposition of the features, this way can make the decomposition results more accurate and effective, but at the same time, it is difficult for the decomposition results to have interpretability.

## 3 PROCESS METHOD

### 3.1 MODEL FRAMEWORK

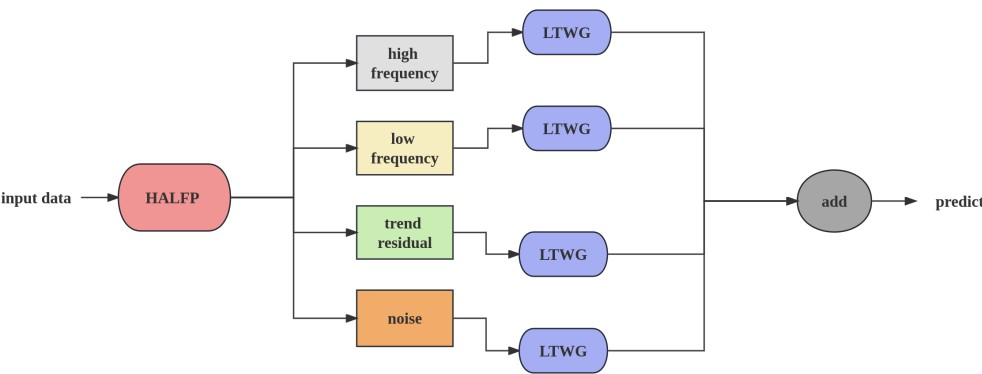

Figure 1: model framework.

As shown in Figure 1, after the input data passes through the HALFP(high and low frequency periods) decomposition module, the data is decomposed into four parts, namely high-frequency periods, low-frequency periods, trend residuals and abnormal data. The HALFP module is a sequence decomposition module we designed by combining the classic time series decomposition algorithm and the mathematical idea of Fourier series. It can decompose time series data into high-frequency

periods, low-frequency periods, trend residuals and abnormal data; We believe that the results of time series forecasting are more determined by periods, and the so-called trend, we believe, is also a period, a low-frequency period. Moreover, we divide trends into long-term trends and short-term trends, which correspond to the above-mentioned low-frequency periods and high-frequency periods. Then, the decomposed data passes through a LTWG module to enhance feature expression and prediction performance. Finally, the LTWG results are added to obtain the final prediction result. The decomposition part is specifically expressed as follows:

$$
\begin{aligned}
X_{\text{trend\_simple}} &= \text{AvgPool}(X)_{kernel} \\
X_{\text{period\_low}} &= \text{fourier\_module}(X_{\text{trend\_simple}}) \\
X_{\text{trend\_residual}} &= X_{\text{trend\_simple}} - X_{\text{period\_low}} \\
X_{\text{period\_high}} &= \text{fourier\_module}(X - X_{\text{trend\_simple}}) \\
X_{noised} &= X_{temp} - X_{\text{period\_high}}
\end{aligned}
\tag{1}
$$

AvgPool is an average pooling layer, we first use the average pooling layer for data smoothing, extract a simple period data $X_{\text{trend\_simple}}$, and then use the Fourier module on $X_{\text{trend\_simple}}$, for low-frequency period feature extraction, to obtain $X_{\text{period\_low}}$. After obtaining $X_{\text{period\_low}}$ low-frequency periodic features, we extract the low-frequency periodic residuals $X_{\text{trend\_residual}}$ by $X_{\text{trend\_simple}} - X_{\text{period\_low}}$. Next, the high-frequency periodic features $X_{\text{period\_high}}$ are extracted on $X - X_{\text{trend\_simple}}$, and finally $X_{noised}$ is obtained by computing $X_{temp} - X_{\text{period\_high}}$, which we consider as noisy data. The decomposition process is shown in Figure 2.

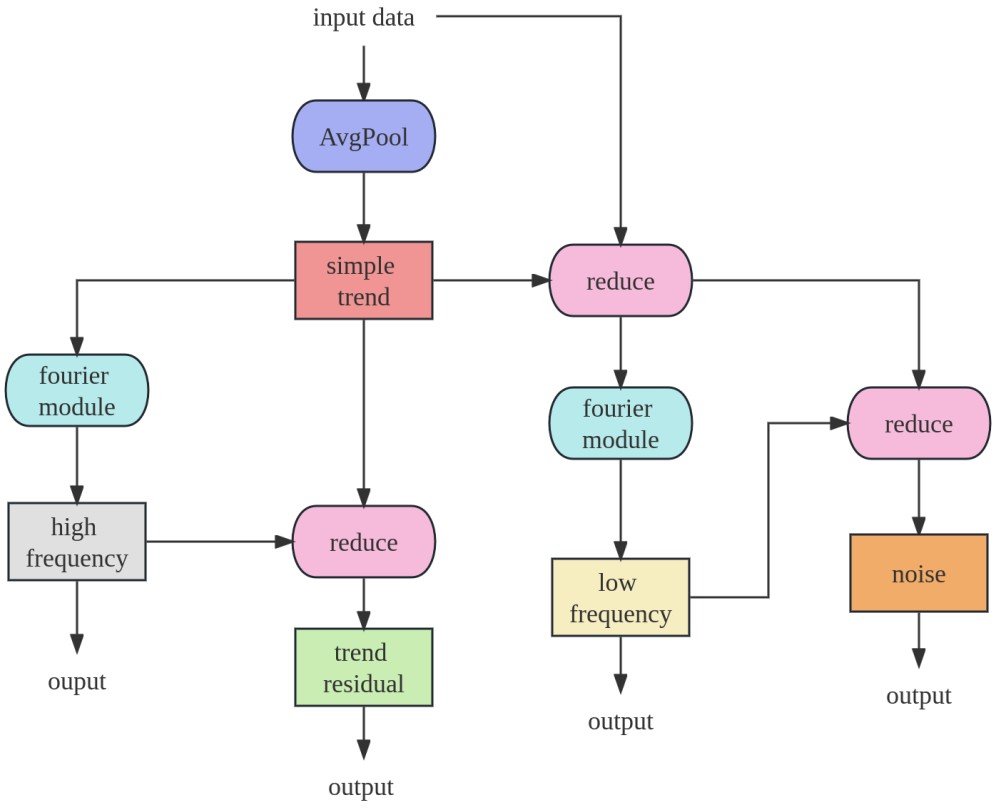

Figure 2: decomposition process.

## 3.2 FOURIER MODULE

We refer to the mathematical idea of Fourier series, that is, a periodic waveform can be represented as a superposition of multiple sine or cosine functions, each sine or cosine function having a different frequency, amplitude and phase. By adjusting the amplitude, frequency, and phase parameters, we can create periodic waveforms of various shapes. The Fourier series can be expressed as follows, where $n$ should tend to infinity:

$$f(x)_{cos} = A_0 + A_1 \cdot \cos(2\pi f_1 x + \phi_1) + A_2 \cdot \cos(2\pi f_2 x + \phi_2) + \ldots + A_n \cdot \cos(2\pi f_n x + \phi_n) \quad (2)$$

$$f(x)_{sin} = A_0 + A_1 \cdot \sin(2\pi f_1 x + \phi_1) + A_2 \cdot \sin(2\pi f_2 x + \phi_2) + \ldots + A_n \cdot \sin(2\pi f_n x + \phi_n) \quad (3)$$

By observing the formula, we find that the Fourier series can be simply implemented through two layers of fully connected layers. The fully connected layer matrix multiplication is expressed as follows:

$$f(x)_{linear} = X \cdot W + b \quad (4)$$

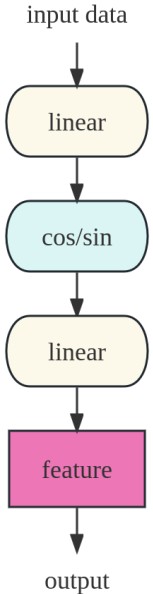

Figure 3: Fourier module.

Where $2\pi f_1 x + \phi_1$, $2\pi f_2 x + \phi_2$, ..., $2\pi f_n x + \phi_n$ in equations (2) and (3) can be obtained through a fully connected layer. Consequently, $\cos(2\pi f_n x + \phi_n)$ and $\sin(2\pi f_n x + \phi_n)$ can be regarded as $\cos(f(X)_{linear})$ and $\sin(f(X)_{linear})$. $A_0$ in formulas (2) and (3) can be regarded as the bias vector $b$ in formula (4), and $A_1, \ldots, A_7, \ldots, A_n$ can be written as the weight matrix $W$ in formula (4). In this way, we can model the Fourier series using two layers of fully connected layers and $cos$ or $sin$. As shown in Figure 3, after the input data passes through a layer of full connection, it is activated by $cos$ or $sin$, and finally passes through a layer of full connection. From this, we complete the reconstruction of a Fourier series deep learning model.

## 3.3 LTWG MODULE

LTWG(local talks with global) module is inspired by MICN(Wang et al. (2022)). MICN first extracts local features of the time series, and then extracts the correlation between all local features to obtain global features. When modeling global features, it does not use attention mechanism with high-complexity, but uses the Isometric Convolution module, thereby reducing computing and memory requirements.

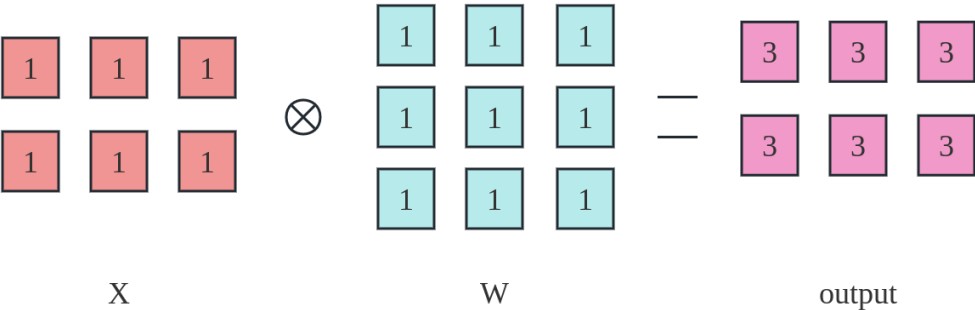

Figure 4: a simple matrix multiplication.

In order to establish the connection between local and global features, different from MICN, our approach is very simple, using only one layer of full connection and one layer of convolution. As shown in the Figure 4, this is a simple matrix multiplication, $W$ is the weight matrix and $X$ is the input data. We can see from the Figure 4 that after matrix multiplication, the input data in the $X$ row direction are each multiplied by a weight and then added together to get 3. In other words, 3 contains all the information in the input data row direction, is information of a global sense.

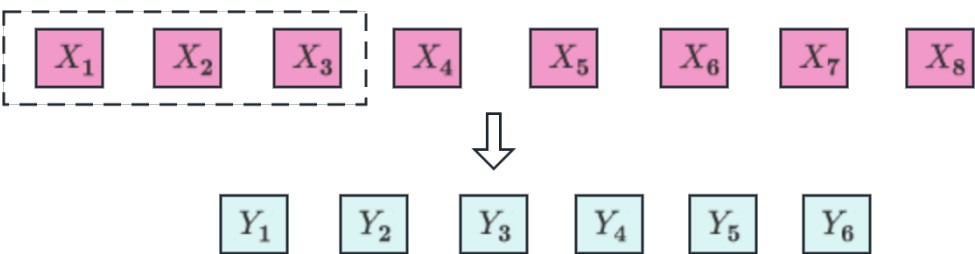

Figure 5: one-dimensional convolution.

In the case of convolution, as shown in Figure 5, this is a one-dimensional convolution with a kernel size of 4, a stride of 1, and zero padding. After the convolution, the size changes from 1x8 to 1x6. Here, $Y_1$ is obtained by multiplying and summing $X_1$, $X_2$, $X_3$ with the convolution kernel weights, and so on for the others. In other words, the features obtained through convolution represent a local feature, and the size of the region it involves is determined by the kernel size.

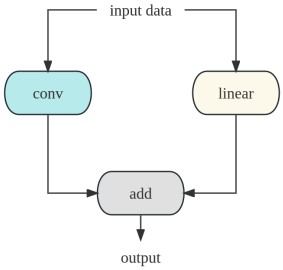

Figure 6: LTWG process.

From the above two points, we can learn that the result of the fully connected layer can be regarded as a global feature, while the result of the convolution can be regarded as a local feature. Therefore our proposed local and global feature fusion approach is very concise, as shown in Figure 6, where the input data is passed through the fully connected and convolutional layers, respectively, and then the two outputs representing the local and global features are added together to establish a link between the local and global features. In this case, in order not to destroy the feature distribution between the data channels, the convolution uses grouped convolution.

# 4 EXPERIMENTS

**Dataset**   In order to evaluate the effectiveness of our proposed model, we have tested it on seven mainstream datasets, which cover most of the scenarios where multivariate time series prediction will be applied in life. The details of the datasets are shown in the Table 1, and the way we divided the datasets is consistent with PatchTST(Nie et al. (2022)), and all the datasets used in this paper are available for download at(Wu et al. (2021)).

| Datasets | Weather | Electricity | ILI | ETTh1 | ETTh2 | ETTm1 | ETTm2 |
|---|---|---|---|---|---|---|---|
| Features | 21 | 321 | 7 | 7 | 7 | 7 | 7 |
| Timesteps | 52696 | 26304 | 966 | 17420 | 17420 | 69680 | 69680 |

Table 1: datasets information

**Baselines**   In order to ensure fair and accurate comparisons, we include state-of-the-art results based on the transformer, CNN-based method as much as possible.

## 4.1 MAIN RESULTS

| Models | | periodNet | | PatchTST/64 | | PatchTST/42 | | best* | |
|---|---|---|---|---|---|---|---|---|---|
| Metric | | MSE | MAE | MSE | MAE | MSE | MAE | MSE | MAE |
| Weather | 96 | **0.149** | **0.195** | 0.149 | 0.198 | 0.152 | 0.199 | **0.149** | 0.198 |
| | 192 | **0.192** | **0.238** | 0.194 | 0.241 | 0.197 | 0.243 | 0.194 | 0.241 |
| | 336 | **0.244** | **0.277** | 0.245 | 0.282 | 0.249 | 0.283 | 0.245 | 0.282 |
| | 720 | 0.321 | **0.329** | 0.314 | 0.334 | 0.320 | 0.335 | **0.314** | 0.334 |
| Electricity | 96 | 0.130 | **0.222** | 0.129 | 0.222 | 0.130 | 0.222 | **0.129** | **0.222** |
| | 192 | **0.147** | **0.239** | 0.147 | 0.240 | 0.148 | 0.240 | **0.147** | 0.240 |
| | 336 | 0.164 | **0.255** | 0.163 | 0.259 | 0.167 | 0.261 | 0.163 | 0.259 |
| | 720 | 0.200 | **0.288** | 0.197 | 0.290 | 0.202 | 0.291 | **0.197** | 0.290 |
| ILI | 24 | **1.311** | **0.726** | 1.319 | 0.754 | 1.522 | 0.814 | 1.319 | 0.754 |
| | 36 | **1.179** | **0.692** | 1.579 | 0.870 | 1.430 | 0.834 | 1.430 | 0.834 |
| | 48 | **1.191** | **0.725** | 1.553 | 0.815 | 1.673 | 0.854 | 1.553 | 0.815 |
| | 60 | **1.442** | 0.802 | 1.470 | 0.788 | 1.529 | 0.862 | 1.470 | **0.788** |
| ETTh1 | 96 | **0.364** | **0.395** | 0.370 | 0.400 | 0.375 | 0.399 | 0.370 | 0.399 |
| | 192 | **0.388** | **0.408** | 0.413 | 0.429 | 0.414 | 0.421 | 0.405 | 0.416 |
| | 336 | **0.404** | **0.423** | 0.422 | 0.440 | 0.431 | 0.436 | 0.422 | 0.436 |
| | 720 | **0.438** | **0.454** | 0.447 | 0.468 | 0.449 | 0.466 | 0.447 | 0.466 |
| ETTh2 | 96 | **0.253** | **0.325** | 0.274 | 0.337 | 0.274 | 0.336 | 0.274 | 0.336 |
| | 192 | **0.294** | **0.354** | 0.341 | 0.382 | 0.339 | 0.379 | 0.339 | 0.379 |
| | 336 | 0.339 | 0.391 | 0.329 | 0.384 | 0.331 | 0.380 | **0.329** | **0.380** |
| | 720 | 0.426 | 0.446 | 0.379 | 0.422 | 0.379 | 0.422 | **0.379** | **0.422** |
| ETTm1 | 96 | 0.291 | **0.340** | 0.293 | 0.346 | 0.290 | 0.342 | **0.290** | 0.342 |
| | 192 | **0.330** | **0.362** | 0.333 | 0.370 | 0.332 | 0.369 | 0.332 | 0.365 |
| | 336 | **0.365** | **0.382** | 0.369 | 0.392 | 0.366 | 0.392 | 0.366 | 0.386 |
| | 720 | 0.425 | **0.415** | 0.416 | 0.420 | 0.420 | 0.424 | **0.416** | 0.420 |
| ETTm2 | 96 | 0.172 | 0.257 | 0.166 | 0.256 | 0.165 | 0.255 | **0.165** | **0.255** |
| | 192 | 0.232 | 0.298 | 0.223 | 0.296 | 0.220 | 0.292 | **0.220** | **0.292** |
| | 336 | 0.281 | 0.332 | 0.274 | 0.329 | 0.278 | 0.329 | **0.274** | **0.329** |
| | 720 | 0.363 | 0.387 | 0.362 | 0.385 | 0.367 | 0.385 | **0.362** | **0.385** |

Figure 7: the best∗ result from patchTSTNie et al. (2022).

**Multivariate results**   As can be seen in Figure 7, our results have achieved state-of-the-art results in all six datasets. The mse of our model is $5.52\%$ lower than the suboptimal result, while on mae, our model is $2.81\%$ lower than the suboptimal result.

## 4.2   ABLATION STUDIES

**Fourier module**   we remove the local and global feature fusion module and replace it with a layer of fully connected, keeping the original Fourier module. Tests are performed on the ili dataset, and the final experimental results are compared with the results of the model without the Fourier module and the local and global feature fusion module, as a way to validate the effectiveness of the designed module. Our comparison results are shown in Table 2.

**LTWG module**   Tested on the ili dataset, we compare the results of the model after removing the Fourier module with the results of the model without the Fourier module and the local and global feature fusion module, and the specific results are shown in Table 2.

| lenght | no LTWG no Fourier | | only Fourier | | only LTWG | | periodNet | |
|---|---|---|---|---|---|---|---|---|
| 24 | 1.624 | 0.846 | 1.331 | 0.734 | 1.529 | 0.759 | 1.311 | 0.726 |
| 36 | 1.610 | 0.837 | 1.243 | 0.716 | 1.590 | 0.801 | 1.179 | 0.692 |
| 48 | 1.593 | 0.849 | 1.282 | 0.742 | 1.553 | 0.800 | 1.191 | 0.725 |
| 60 | 1.750 | 0.901 | 1.490 | 0.814 | 1.735 | 0.867 | 1.442 | 0.802 |

Table 2: datasets information

## 5   CONCLUSIONS

From our comparison results and ablation experiments, we can learn that both our proposed Fourier module and LTWG module are effective in improving the timing prediction accuracy with fewer parameters.

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
