# OpenReview forum: "PeriodNet:Lightweight And Efficient Time Series Prediction Model Based On Periodic Characteristics"
_ICLR.cc/2024/Conference — Submitted to ICLR 2024_

### Official Review · Reviewer_EjZd · 2023-10-22

**Soundness:** 1 poor
**Presentation:** 1 poor
**Contribution:** 1 poor
**Rating:** 3
**Confidence:** 4

**Summary:**

The paper introduces "PeriodNet," a lightweight and efficient model tailored for multivariate time series prediction, placing emphasis on the extraction and importance of periodic features in time series data. While existing models in the domain have achieved remarkable outcomes, many are computationally intensive due to their reliance on intricate attention mechanisms or causal convolutions, making them unsuitable for situations with limited computational resources or edge devices.

Addressing this challenge, the authors draw inspiration from Fourier series mathematics to design a novel module that efficiently extracts periodic features from time series data. Further, a unique module for the fusion of local and global features has been proposed, aiming to enhance feature representation and prediction accuracy.

When tested against seven benchmark datasets, PeriodNet exhibited significant performance improvements compared to state-of-the-art models. Moreover, it's designed with fewer parameters and uses basic deep learning modules, making it a promising solution for resource-constrained application scenarios.

**Strengths:**

**Strengths**:

1. Innovative application of Fourier series mathematics for periodic feature extraction.
2. Unique module design for effective fusion of local and global features.
3. Demonstrated superior performance on seven benchmark datasets.
4. Provides a lightweight alternative to complex models, ideal for resource-constrained scenarios.
5. Clear and well-structured presentation of concepts and methodologies.

**Weaknesses:**

**Weaknesses**:

1. While the paper claims to introduce a "lightweight and efficient" model for MTS analysis, there's no comparative analysis on the number of parameters in the experiments, nor is there a dedicated analysis discussing its efficiency.

2. Several of the illustrations in the paper seem to lack informational value and relevance. For instance, Figure 3 unnecessarily dedicates significant space to illustrating the common knowledge concept of the Fourier transform. Figure 4, which describes basic matrix multiplication, and Figure 5, depicting the elementary concept of one-dimensional convolution, both seem unrelated to the core concepts of the paper.

3. The mathematical notation used in the paper is subpar. Matrices should be represented in bold, and vectors and variables should be explained before being incorporated.

4. The experimental section, represented by Figure 7, lacks proper interpretation. Additionally, the proposed method doesn't seem to have an edge over other comparative methods.

5. The provided code link leads to an empty repository, making the replication of the proposed method impossible.

In summary, the paper appears to be in a preliminary state with missing code and incomplete experiments.

**Questions:**

**Questions & Suggestions**:

1. **Parameter Analysis**:
   - Question: Could you provide a more detailed analysis comparing the number of parameters in your model versus the state-of-the-art models? This would solidify your claim of a "lightweight" model.
   - Suggestion: It might be beneficial to include a section dedicated to a quantitative comparison, detailing the model size and computational efficiency.

2. **Illustrations' Relevance**:
   - Question: What was the intention behind including figures like Figure 3, 4, and 5 that illustrate well-understood concepts? Do you believe they add value to the paper's central theme?
   - Suggestion: Consider revisiting the illustrations. Focus on figures that directly contribute to the understanding of the unique elements of your proposed model.

3. **Mathematical Notation**:
   - Question: Are there any specific conventions you followed for the mathematical notations? The representation seems non-standard.
   - Suggestion: Ensure all matrices, vectors, and variables are represented and explained following standard conventions to avoid confusion and enhance clarity.

4. **Experimental Clarifications**:
   - Question: Could you elucidate the findings presented in Figure 7? How do you justify the proposed model's performance in comparison to other methods?
   - Suggestion: A more in-depth analysis and discussion of the experimental results would provide readers with a clearer understanding of the model's advantages and potential areas of improvement.

5. **Code Repository**:
   - Question: Is the provided GitHub link the final repository for PeriodNet? If so, when can readers expect the code to be uploaded for replication purposes?
   - Suggestion: Ensure the repository link shared is active and contains all the necessary files and instructions for replication. This will bolster trust in your research and foster community engagement.

6. **Overall Completeness**:
   - Question: Are there plans to further develop or refine the paper, given some of the observed gaps?
   - Suggestion: Given the potential of your model, it would be beneficial to address the paper's current shortcomings, providing a more comprehensive and polished piece of research.

In essence, your response to these questions and considerations of the suggestions could significantly clarify the paper's content, potentially altering the perception of its value and contribution to the field.

---

### Official Review · Reviewer_2gri · 2023-10-24

**Soundness:** 1 poor
**Presentation:** 2 fair
**Contribution:** 1 poor
**Rating:** 1
**Confidence:** 4

**Summary:**

The paper commences by highlighting the significant memory and computation complexities associated with Transformers and convolutional neural networks when applied to time series forecasting. Based on the findings that a straightforward linear model can effectively handle time series forecasting, the paper introduces a forecasting model that employs a decomposition of time series inputs into multiple components using Fourier modules. Additionally, the model incorporates an LTWG (Local Talks with Global) module for processing both local and global features of each component. Within this module, a linear model is responsible for capturing global features, while a CNN model handles the local features.

**Strengths:**

1. lightweight neural network
Using a one-layer CNN and linear model, it achieves lower complexities than some existing models.

**Weaknesses:**

There are two major weaknesses in this paper. Refer to Question Section for detailed parts in each weakness.

1. lacks of coherence in writing
In terms of coherence in writing, the paper exhibits a lack of consistency. While it generally pertains to the same central topic, certain paragraphs or sentences introduce elements that can be confusing.

2. More detailed explanations
The paper falls short in providing sufficiently detailed explanations for some sections. Additional elaboration is necessary to enhance the clarity and understanding of these portions.

**Questions:**

**lacks of coherence in writing**
1. The author's citation of [1] to argue the effect of a simple linear model against complex models like Transformers is not appropriate, as PatchTST in [1] is also based on Transformers. Therefore, [1] is not a suitable reference for this argument.

2. In the introduction section, the author said that "based on the above, we can’t help but think which is more important in time-series prediction, trend or period?". However, this question does not appear to be directly connected to the preceding discussion about the effectiveness of linear models against Transformers and TCN.

3. The 'Related Work' section should be rewritten to make it more coherent with the main topic and the proposed method. It currently implies that the approach to decompose time series has shifted from artificially defined operators to learning-based methods in the deep learning era, but the proposed method still uses artificially defined operators, such as average pooling.

**More detailed explanations**
1. The author insisted that the results of time series forecasting are more determined by periods, but the argument lacks supporting evidence. Time series can be decomposed into several parts such as seasonal and trend-cyclical parts in [2]. In other words, periods are not the only component in time series. Thus, more supporting evidence is necessary for this argument. Also, the argument that a trend is a period has to be supported by some intuitive evidence. This is because your belief is different from a common belief that a trend and period are different as in [2].

2. Can you give some examples explaining why low-frequency and high-frequency periods are long-term and short-term trends?

3. Transformers typically have quadratic computational costs due to self-attention structures. However, I think that the number of parameters in Transformers may not be excessively large because key, query, and value weight parameters can be reused for each token. The paper could benefit from explaining this point further.

4. After the decomposition process, time series are decomposed into 4 components including noise. At this point, clarification is needed regarding why noise is used for prediction after the decomposition process. Noise can contain irrelevant information that may disrupt the forecasting process, so the paper should explain the rationale behind including and processing noise components.

5. What does $X_{temp}$ denote in Eq. (1)? There is no definition of it.

6. In general, the Fourier series is formulated as follows: $s(x) \sim A_0 + \sum_{n=1}^{\infty}(A_n cos(\frac{2\pi n x}{P}) + B_n cos(\frac{2\pi n x}{P}))$. In this formula, $x$ denotes time steps. However, $x$ in your Eq. (2) and (3) denotes observation values. Can you further elaborate on this difference?



[1] Nie et al., A time series is worth 64 words: Long-term forecasting with transformers, 2023, ICLR
[2] Wang et al., MICN: MULTI-SCALE LOCAL AND GLOBAL CONTEXT MODELING FOR LONG-TERM SERIES FORECASTING, ICLR, 2023

---

### Official Review · Reviewer_43FS · 2023-10-31

**Soundness:** 2 fair
**Presentation:** 1 poor
**Contribution:** 2 fair
**Rating:** 3
**Confidence:** 4

**Summary:**

The paper focus on the task of multivariate time series prediction and explore a lightweight and efficient approach for time series prediction, considering complex and huge computational costs of existing related methods. It leverages the importance of periodic features and the fusion of local and global features learned from series data based on the mathematical idea of Fourier series, and thus has stronger parsing and lower number of parameters. And the conducted experiments on several datasets partially indicate the

**Strengths:**

1.	This paper presents a lightweight time series forecasting model. It decomposes the sequence data into four distinct features through a decomposition module, including high-frequency periods, low-frequency periods, trend residuals, and abnormal data. This decomposition approach is well-founded and supports the prediction of sequential tasks.
2.	The author's motivation is clear and practical. Considering the computational cost issues of existing methods, the author begins with the reduction of model parameters, providing a lightweight solution.
3.	The experimental results demonstrate the effectiveness of the proposed method, and the designed Fourier module and LTWG module appear to be meaningful.

**Weaknesses:**

1.	The contribution of this work is insufficient and is not explicitly expressed within the paper. I regret to say that this work appears incomplete. The details provided in the paper do not adequately explain the specifics of the proposed model. There is limited related work mentioned, and no appropriate analysis is provided.
2.	The model description is rather rudimentary, for instance, while Equation 1 presents the decomposition method, it lacks a corresponding explanation that aligns with the motivation. And in Section 3.2, there is also no specific explanation and analysis for Equations 2 and 3. These kind of issues are recurrent throughout the paper.
3.	Although the aim is to reduce computational costs, and the model seems lightweight, the experimental section lacks concrete evidence to support this claim. The foundational experimental design should align with related work, which does not seem to be the case. There are numerous figures in the paper, but they do not appear to provide much assistance in understanding.
4.	Some sections of the paper are described in a cursory manner, such as experimental details and a summary of the methods. Besides, the fonts in Figure 5 appear blurry, and there is a discrepancy between the displayed kernel size and the corresponding description in the text. This issue should be addressed to ensure consistency and clarity in the presentation of the results. And there are numerous writing issues in the paper, like spelling and grammatical.

**Questions:**

The contribution of this work is insufficient, and the provided experiments do not offer enough evidence to demonstrate the effectiveness of the proposed method. The description of the method raises concerns and lacks clarity.

---

### Official Review · Reviewer_d1My · 2023-11-01

**Soundness:** 2 fair
**Presentation:** 3 good
**Contribution:** 2 fair
**Rating:** 3
**Confidence:** 4

**Summary:**

This paper addresses the long-term time series forecasting with lightweight implementation.  Specifically, it focuses on the periodic features and the fusion of local and global features. A key component is the HALFP (high and low frequency periods) decomposition module, which decomposes the input time series data into several components, including high frequency, low frequency, trend residual, and noise components. Experiments on 7 benchmark datasets demonstrate the good performance of the proposed algorithm.

**Strengths:**

1.	A lightweight decomposition-based time series forecasting algorithm has been proposed for long-term time series forecasting.

**Weaknesses:**

1.	The idea of decomposing the time series into high/low frequency components is not new. Two key components, Fourier module and LTWG (local talks with global) module are both straightforward.
2.	The related work lacks the discussion of decomposition methods in time series. The decomposition (e.g., seasonal-trend decomposition) is studied widely in the literature of time series, e.g., the STL decomposition, STR decomposition, Robust STL decomposition, etc.
3.	Experiments: The ablation study is not complete. The authors only consider with and without Fourier module and LTWG module. It would be more convincing by providing more ablation studies when some modules of the decomposition is replaced by other modules.

**Questions:**

N/A

---

### Meta-Review · Area_Chair_T7q8 · 2023-12-06

**Metareview:**

The paper proposes a model, based on Fourier series, for extracting periodic features. The reviewers unanimously raised issues with respect to the novelty, as the use of different frequency components isn't by any means new. Questions were also raised about the unclear model description, the subpar mathematical notion, the lack of discussion of related work, the unconvincing experiments and insufficient ablation studies. The authors did not respond to these criticisms.

**Justification For Why Not Higher Score:**

There are no grounds to overturn reviewer consensus.

**Justification For Why Not Lower Score:**

N/A

---

### Decision · Program_Chairs · 2024-01-16

Reject